# A mobile-optimized artificial intelligence system for gestational age and fetal malpresentation assessment

Ryan G. Gomes [1,6✉], Bellington Vwalika[2,3,6], Chace Lee[1,6], Angelica Willis[1,6], Marcin Sieniek[1], Joan T. Price[3,4], Christina Chen[1], Margaret P. Kasaro[2,4], James A. Taylor[1], Elizabeth M. Stringer[3], Scott Mayer McKinney [1], Ntazana Sindano[4], George E. Dahl [5], William Goodnight III[2], Justin Gilmer[5], Benjamin H. Chi[3,4], Charles Lau [1], Terry Spitz [1], T. Saensuksopa[1], Kris Liu[1], Tiya Tiyasirichokchai[1], Jonny Wong[1], Rory Pilgrim[1], Akib Uddin[1✉], Greg Corrado[1], Lily Peng[1], Katherine Chou[1], Daniel Tse [1✉], Jeffrey S. A. Stringer [3,4,7] & Shravya Shetty [1,7✉]

## Abstract

**Background** Fetal ultrasound is an important component of antenatal care, but shortage of adequately trained healthcare workers has limited its adoption in low-to-middle-income countries. This study investigated the use of artificial intelligence for fetal ultrasound in under-resourced settings.

**Methods** Blind sweep ultrasounds, consisting of six freehand ultrasound sweeps, were collected by sonographers in the USA and Zambia, and novice operators in Zambia. We developed artificial intelligence (AI) models that used blind sweeps to predict gestational age (GA) and fetal malpresentation. AI GA estimates and standard fetal biometry estimates were compared to a previously established ground truth, and evaluated for difference in absolute error. Fetal malpresentation (non-cephalic vs cephalic) was compared to sonographer assessment. On-device AI model run-times were benchmarked on Android mobile phones.

**Results** Here we show that GA estimation accuracy of the AI model is non-inferior to standard fetal biometry estimates (error difference $-1.4 \pm 4.5$ days, 95% CI $-1.8$, $-0.9$, $n = 406$). Non-inferiority is maintained when blind sweeps are acquired by novice operators performing only two of six sweep motion types. Fetal malpresentation AUC-ROC is 0.977 (95% CI, 0.949, 1.00, $n = 613$), sonographers and novices have similar AUC-ROC. Software run-times on mobile phones for both diagnostic models are less than 3 s after completion of a sweep.

**Conclusions** The gestational age model is non-inferior to the clinical standard and the fetal malpresentation model has high AUC-ROCs across operators and devices. Our AI models are able to run on-device, without internet connectivity, and provide feedback scores to assist in upleveling the capabilities of lightly trained ultrasound operators in low resource settings.

## Plain language summary

Despite considerable progress in maternal healthcare, maternal and perinatal deaths remain high in low-to-middle income countries. Fetal ultrasound is an important component of antenatal care, but shortage of adequately trained healthcare workers has limited its adoption. We developed and validated an automated system that enables lightly-trained community healthcare providers to conduct ultrasound examinations. Our approach uses artificial intelligence to automatically interpret ultrasound video acquired by sweeping an ultrasound device across the patient's abdomen, a procedure that can easily be taught to non-experts. Our system consists of a low cost battery-powered ultrasound device and a smartphone, and can operate without internet connectivity or other infrastructure, making it suitable for deployment in low-resourced settings. The accuracy of our method is on par with existing clinical standards. Our approach has the potential to improve access to ultrasound in low-resource settings.

[1] Google Health, Palo Alto, CA, USA. [2] Department of Obstetrics and Gynaecology, University of Zambia School of Medicine, Lusaka, Zambia. [3] Department of Obstetrics and Gynecology, University of North Carolina School of Medicine, Chapel Hill, NC, USA. [4] UNC Global Projects—Zambia, LLC, Lusaka, Zambia. [5] Google Research, Mountain View, CA, USA. [6] These authors contributed equally: Ryan G. Gomes, Bellington Vwalika, Chace Lee, Angelica Willis. [7] These authors jointly supervised this work: Jeffrey S. A. Stringer, Shravya Shetty. ✉email: ryangomes@google.com; akibu@google.com; tsed@google.com; sshetty@google.com

Despite considerable progress in maternal healthcare in recent decades, maternal and perinatal deaths remain high with 295,000 maternal deaths during and following pregnancy and 2.4 million neonatal deaths each year. The majority of these deaths occur in low-to-middle-income countries (LMICs)[1–3]. The lack of antenatal care and limited access to facilities that can provide lifesaving treatment for the mother, fetus and newborn contribute to inequities in quality of care and outcomes in these regions[4,5].

Obstetric ultrasound is an important component of quality antenatal care. The WHO recommends one routine early ultrasound scan for all pregnant women, but up to 50% of women in developing countries receive no ultrasound screening during pregnancy[6]. Fetal ultrasounds can be used to estimate gestational age (GA), which is critical in scheduling and planning for screening tests throughout pregnancy and interventions for pregnancy complications such as preeclampsia and preterm labor. Fetal ultrasounds later in pregnancy can also be used to diagnose fetal malpresentation, which affects up to 3–4% of pregnancies at term and is associated with trauma-related injury during birth, perinatal mortality, and maternal morbidity[7–11].

Though ultrasound devices have traditionally been costly, the recent commercial availability of low-cost, battery-powered handheld devices could greatly expand access[12–14]. However, current ultrasound training programs require months of supervised evaluation as well as indefinite continuing education visits for quality assurance[13–19]. GA estimation and diagnosis of fetal malpresentation require expert interpretation of anatomical imagery during the ultrasound acquisition process. GA estimation via clinical standard biometry[20] requires expertly locating fetal anatomical structures and manually measuring their physical sizes in precisely collected images (head circumference, abdominal circumference, femur length, among others). To address these barriers, prior studies have introduced a protocol where fetal ultrasounds can be acquired by minimally trained operators via a "blind sweep" protocol, consisting of six predefined freehand sweeps over the abdomen[21–27]. While blind-sweep protocols simplify the ultrasound acquisition process, new methods are required for interpreting the resulting imagery. AI-based interpretation may provide a promising direction for generating automated clinical estimates from blind-sweep video sequences.

In this study, we used two prospectively collected fetal ultrasound datasets to estimate gestational age and fetal malpresentation while demonstrating key considerations for use by novice users in LMICs: (a) validating that it is possible to build blind-sweep GA and fetal malpresentation models that run in real-time on mobile devices; (b) evaluating generalization of these models to minimally trained ultrasound operators and low-cost ultrasound devices; (c) describing a modified 2-sweep blind-sweep protocol to simplify novice acquisition; (d) adding feedback scores to provide real-time information on sweep quality.

## Methods

**Blind-sweep procedure**. Blind-sweep ultrasounds consisted of a fixed number of predefined freehand ultrasound sweeps over the gravid abdomen. Certified sonographers completed up to 15 sweeps. Novice operators ("novices"), with 8 h of blind-sweep ultrasound acquisition training, completed six sweeps. Evaluation of both sonographers and novices was limited to a set of six sweeps—three vertical and three horizontal sweeps (Fig. 1b).

**Fetal age machine learning initiative (FAMLI) and novice user study datasets**. Data were analyzed from the Fetal Age Machine Learning Initiative cohort, which collected ultrasound data from study sites at Chapel Hill, NC (USA), and the Novice User Study collected from Lusaka, Zambia (Fig. 1a)[27]. The goal of this prospectively collected dataset was to enable the development of technology to estimate gestational age[28]. Data collection occurred between September 2018 and June 2021. All study participants provided written informed consent, and the research was approved by the UNC institutional review board (IRB #18-1848) and the biomedical research ethics committee at the University of Zambia. Blind-sweep data were collected with standard ultrasound devices (SonoSite M-Turbo or GE Voluson) as well as a low-cost portable ultrasound device (ButterflyIQ). Studies included standard clinical assessments of GA[20] and fetal malpresentation performed by a trained sonographer using a standard ultrasound device.

**Algorithm development**. We developed two deep learning neural network models to predict GA and fetal malpresentation. Our models generated diagnostic predictions directly from ultrasound video: sequences of image pixel values were the input and an estimate of the clinical quantity of interest was the output. The GA model produced an estimate of age, measured in days, for each blind-sweep video sequence. The GA model additionally provided an estimate of its confidence in the estimate for a given video sequence. No intermediate fetal biometric measurements were required during training or generated during inference. The fetal malpresentation model predicted a probability score between 0.0 and 1.0 for whether the fetus is in noncephalic presentation. See Supplementary Materials for a technical discussion and details regarding model development.

In the USA, the ground truth GA was determined for each participant based on the "best obstetric estimate," as part of routine clinical care, using procedures recommended by the American College of Obstetricians and Gynecologists (ACOG)[29]. The best obstetric estimate combines information from the last menstrual period (LMP), GA derived from assisted reproductive technology (if applicable), and fetal ultrasound anatomic measurements. In Zambia, only the first fetal ultrasound was used to determine the ground truth GA as the LMP in this setting was considered less reliable as patients often presented for care later in pregnancy.

The GA model was trained on sonographer-acquired blind sweeps (up to 15 sweeps per patient) as well as sonographer-acquired "fly-to" videos that capture five to ten seconds before the sonographer has acquired standard fetal biometry images. The fetal malpresentation model was only trained on blind sweeps. For each training set case, fetal malpresentation was specified as one of four possible values by a sonographer (cephalic, breech, transverse, oblique), and dichotomized to "cephalic" vs "noncephalic". This dichotomization is clinically justified since cephalic cases are considered normal while all noncephalic cases require further medical attention.

Our analysis cohort included all pregnant women in the FAMLI and Novice User Study datasets who had the necessary ground truth information for gestational age and fetal presentation from September 2018 to January 2021. Study participants were assigned at random to one of three dataset splits: train, tune, or test. We used the following proportions: 60% train/20% tune/20% test for study participants who did not receive novice sweeps, and 10% tune/90% test for participants who received novice sweeps. The tuning set was used for optimizing machine learning training hyperparameters and selecting a classification threshold probability for the fetal malpresentation model. This threshold was chosen to yield equal noncephalic specificity and sensitivity on the tuning set, blinded to the test sets. None of the blind-sweep data collected by the novices were used for training.

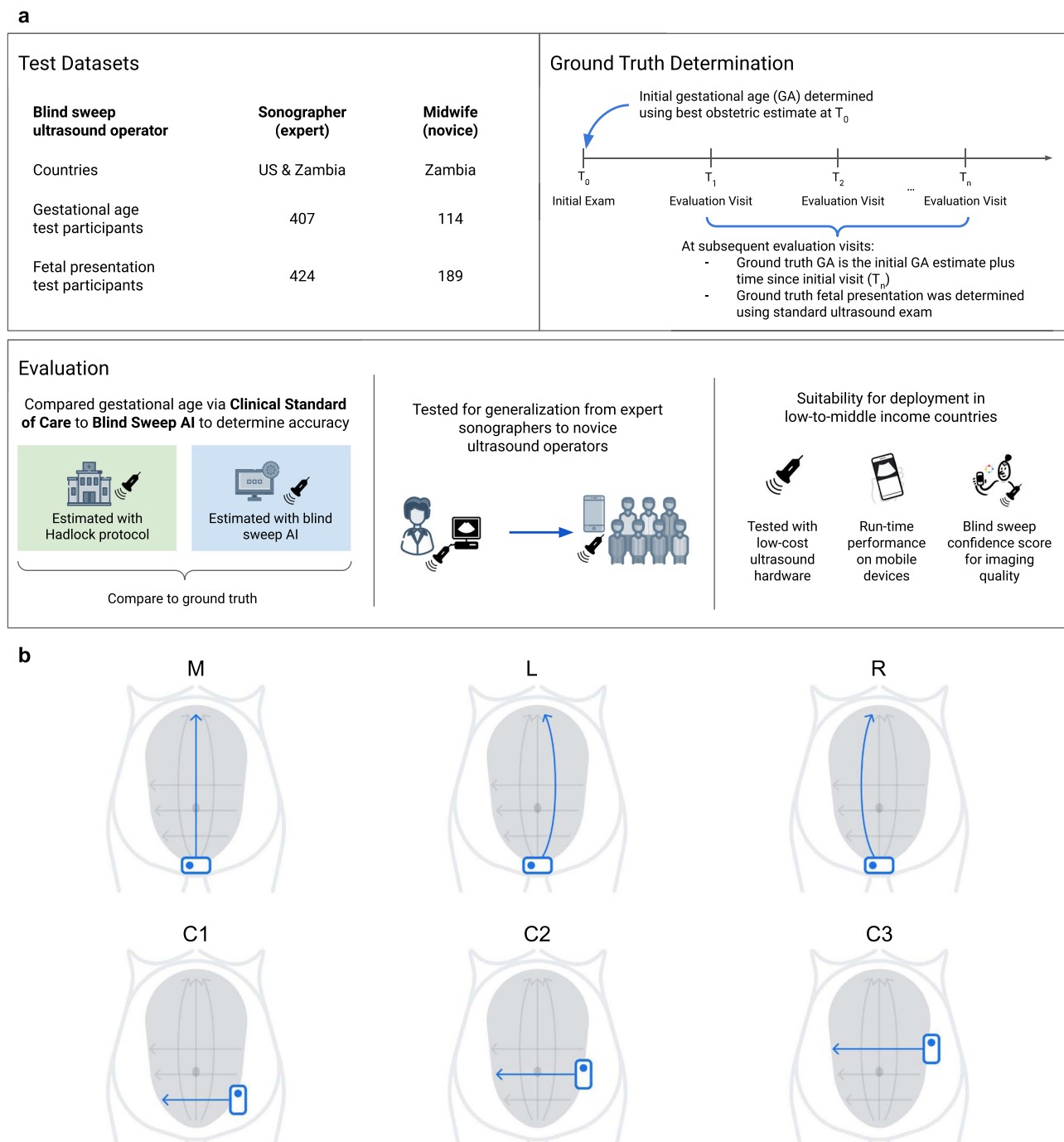

**Fig. 1 Development of an artificial intelligence system to acquire and interpret blind-sweep ultrasound for antenatal diagnostics. a** Datasets were curated from sites in Zambia and the USA and include ultrasound acquired by sonographers and midwives. Ground truth for gestational age was derived from the initial exam as part of clinical practice. An artificial intelligence (AI) system was trained to identify gestational age and fetal malpresentation and was evaluated by comparing the accuracy of AI predictions with the accuracy of clinical standard procedures. The AI system was developed using only sonographer blind-sweep data, and its generalization to novice users was tested on midwife data. Design of the AI system considered suitability for deployment in low-to-middle-income countries in three ways: first, the system interpreted ultrasound from low-cost portable ultrasound devices; second, near real-time interpretation is available offline on mobile phone devices; and finally, the AI system produces feedback scores that can be used to provide feedback to users. **b** Blind-sweep ultrasound acquisition procedure. The procedure can be performed by novices with a few hours of ultrasound training. While the complete protocol involves six sweeps, a set of two sweeps (M and R) were found to be sufficient for maintaining the accuracy of gestational age estimation.

Cases consisted of multiple blind-sweep videos, and our models generated predictions independently for each video sequence within the case. For the GA model, each blind sweep was divided into multiple video sequences. For the fetal malpresentation model, video sequences corresponded to a single complete blind sweep. We then aggregated the predictions to generate a single case-level estimate for either GA or fetal malpresentation (described further in the Mobile Device Inference section in supplementary materials).

**Table 1 Gestational age estimation.**

| | Sweeps collected by sonographers | | Sweeps collected by novices | |
|---|---|---|---|---|
| | Standard ultrasound device | Low-cost handheld device | Standard ultrasound device | Low-cost handheld device |
| Number | 406 | 104 | 112 | 56 |
| Blind-sweep MAE ± sd (days) | 3.8 ± 3.6 | 3.3 ± 2.8 | 4.4 ± 3.5 | 5.0 ± 4.0 |
| Standard fetal biometry estimates MAE ± sd (days) | 5.2 ± 4.6 | 3.8 ± 3.6 | 4.8 ± 3.7 | 4.7 ± 4.0 |
| Blind sweep—standard fetal biometry mean difference ± sd (days) | −1.4 ± 4.5 | −0.6 ± 3.8 | −0.4 ± 4.8 | 0.4 ± 5.1 |
| MAE difference 95% CI (days) | −1.8, −0.9 | −1.3, 0.1 | −1.3, 0.5 | −1.0, 1.7 |
| Blind sweep ME ± sd (days) | −0.9 ± 5.3 | 0.4 ± 4.4 | −1.5 ± 5.5 | −3.8 ± 5.4 |
| Standard fetal biometry estimates ME ± sd (days) | −1.4 ± 7.0 | −0.25 ± 5.4 | −2.6 ± 5.3 | −3.4 ± 5.2 |
| Reduced blind-sweep protocol MAE ± sd (days) | 4.0 ± 3.7 | 3.5 ± 3.0 | 4.5 ± 3.5 | 5.1 ± 4.2 |

Mean absolute error (MAE) and mean error (ME) between gestational age (GA) estimated using the blind-sweep procedure and ground truth, and the MAE and ME between the GA estimated using the standard fetal biometry ultrasound procedure and ground truth. One visit by each participant eligible for each subgroup was selected at random. The reduced blind-sweep protocol (last row) included only two blind sweeps. All other blind-sweep results used a set of six blind sweeps per patient visit. All fetal biometry GA estimates were collected by expert sonographers using standard ultrasound devices.

**Evaluation**. The evaluation was performed on the FAMLI (sonographer-acquired) and Novice User Study (novice-acquired) datasets. Test sets consisted of patients independent of those used for AI development (Fig. 1a). For our GA model evaluation, the primary FAMLI test set comprised 407 women in 657 study visits in the USA. A second test set, "Novice User Study" included 114 participants in 140 study visits in Zambia. Novice blind-sweep studies were exclusively performed at Zambian sites. Sweeps collected with standard ultrasound devices were available for 406 of 407 participants in the sonographer-acquired test set, and 112 of 114 participants in the novice-acquired test set. Sweeps collected with the low-cost device were available for 104 of 407 participants in the sonographer-acquired test set, and 56 of 114 participants in the novice-acquired test set. Analyzable data from the low-cost device became available later during the study, and this group of patients is representative of the full patient set. We randomly selected one study visit per patient for each analysis group to avoid combining correlated measurements from the same patient. For our fetal malpresentation model, the test set included 613 patients from the sonographer-acquired and novice-acquired datasets, resulting in 65 instances of noncephalic presentation (10.6%). For each patient, the last study visit of the third trimester was included. Of note, there are more patients in the malpresentation model test set since the ground truth is not dependent on a prior visit. The disposition of study participants are summarized in STARD diagrams (Supplementary Fig. 1) and Supplementary Table 1.

**Reporting summary**. Further information on research design is available in the Nature Research Reporting Summary linked to this article.

## Results

**Mobile-device-optimized AI gestational age and fetal malpresentation estimation**. We calculated the mean difference in absolute error between the GA model estimate and estimated gestational age as determined by standard fetal biometry measurements using imaging from traditional ultrasound devices operated by sonographers[20]. The reference ground truth GA was established based on an initial patient visit as described above in Methods. When conducting pairwise statistical comparisons between blind sweep and standard fetal biometry absolute errors, we established an a priori criterion for non-inferiority which was confirmed if the blind-sweep mean absolute error (MAE) was less

than 1.0 day greater than the standard fetal biometry's MAE. Statistical estimates and comparisons were computed after randomly selecting one study visit per patient for each analysis group, to avoid combining correlated measurements from the same patient.

We conducted a supplemental analysis of GA model prediction error with mixed effects regression on all test data, combining sonographer-acquired and novice-acquired test sets. Fixed effect terms accounted for the ground truth GA, the type of ultrasound machine used (standard vs. low cost), and the training level of the ultrasound operator (sonographer vs. novice). All patient studies were included in the analysis, and random effects terms accounted for intra-patient and intra-study effects.

GA analysis results are summarized in Table 1. The MAE for the GA model estimate with blind sweeps collected by sonographers using standard ultrasound devices was significantly lower than the MAE for the standard fetal biometry estimates (mean difference −1.4 ± 4.5 days, 95% CI −1.8, −0.9 days). There was a trend toward increasing error for a blind sweep and standard fetal biometry procedures with the gestational week (Fig. 2a).

The accuracy of the fetal malpresentation model for predicting noncephalic fetal presentation from third-trimester blind sweeps was assessed using a reference standard determined by sonographers equipped with traditional ultrasound imagery (described above). We selected the latest study visit in the third trimester for each patient. Data from sweeps performed by the sonographers and novices were analyzed separately. We evaluated the fetal malpresentation model's area under the receiver operating curve (AUC-ROC) on the test set in addition to noncephalic sensitivity and specificity.

The fetal malpresentation model attained an AUC-ROC of 0.977 (95% CI 0.949, 1.00), sensitivity of 0.938 (95% CI 0.848, 0.983), and specificity of 0.973 (95% CI 0.955, 0.985) (Table 2 and Fig. 3).

**Generalization of GA and malpresentation estimation to novices**. Our models were trained on up to 15 blind sweeps per study performed by sonographers. No novice-acquired blind sweeps were used to train our models. We assessed GA model generalization to blind sweeps performed by novice operators that performed six sweeps. We compared the MAE between novice-performed blind-sweep AI estimates and the standard fetal biometry. For the malpresentation model, we reported the

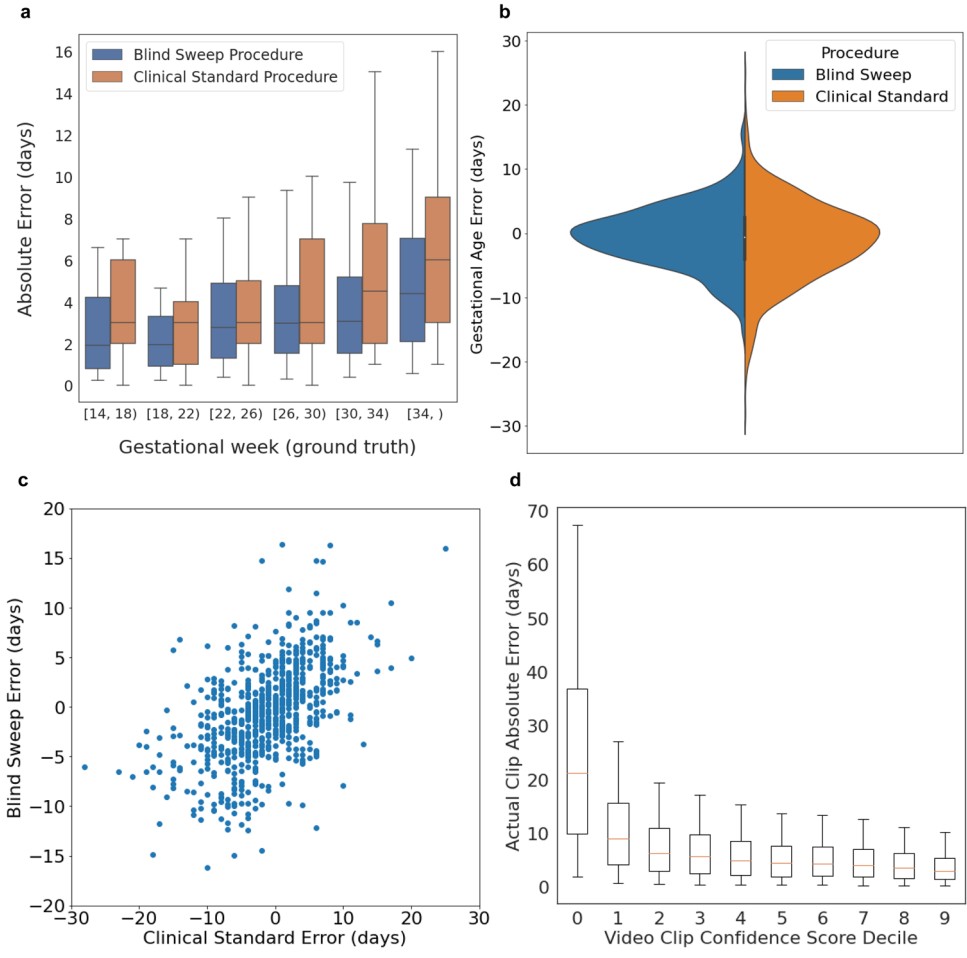

**Fig. 2 Gestational age estimation.** $n = 407$ study participants, blind sweeps performed by expert sonographers. **a** Blind-sweep procedure and standard fetal biometry procedure absolute error versus ground truth gestational age (4-week windows). Box indicates 25th, 50th, and 75th percentile absolute error, and whiskers indicate 5th and 95th percentile absolute error. **b** Error distributions for blind-sweep procedure and standard fetal biometry procedure. **c** Paired errors for a blind sweep and standard fetal biometry estimates in the same study visit. The errors of the two methods exhibit correlation, but the worst-case errors for the blind-sweep procedure have a lower magnitude than the standard fetal biometry method. **d** Video sequence feedback-score calibration on the test sets. The realized model estimation error on held-out video sequences decreases as the model's feedback score increases. A thresholded feedback score may be used as a user feedback signal to redo low-quality blind sweeps. Box indicates 25th, 50th, and 75th percentile of absolute errors, and whiskers indicate the 5th and 95th percentile absolute error.

AUC-ROC for blind sweeps performed by novices, along with the sensitivity and specificity at the same operating point used for evaluating blind sweeps performed by sonographers.

In this novice-acquired dataset, the difference in MAE between blind-sweep AI estimates and the standard fetal biometry was −0.6 days (95% CI −1.7, 0.5), indicating that sweeps performed by novice operators provide a non-inferior GA estimate compared to the standard fetal biometry. Table 1 provides novice blind-sweep performance analyzed by ultrasound device type. The mixed effects regression error analysis did not indicate a significant association between GA error magnitude and the type of operator conducting the blind sweep ($P = 0.119$).

Fetal malpresentation using novice-acquired blind sweeps was compared to the sonographer's determination on 189 participants (21 malpresentations), and AUC-ROC was 0.992 (95% CI 0.983, 1.0). On the preselected operating point, sensitivity was 1.0 (95% CI 0.839, 1.0) and specificity was 0.952 (95% CI 0.908, 0.979).

**Performance of low-cost ultrasound device in GA and fetal malpresentation estimation**. GA model estimation using blind sweeps acquired with the low-cost ultrasound device were compared against the clinical standard on the combined

novice-acquired and sonographer-acquired test sets. We used the same a priori criterion for non-inferiority as described above, 1.0 day. For the malpresentation model, we reported AUC-ROC by type of ultrasound device along with sensitivity and specificity at the same operating point discussed above.

GA model estimation using blind sweeps acquired with the low-cost ultrasound device were compared against the standard fetal biometry estimates on the combined novice-acquired and sonographer-acquired test sets. The blind-sweep AI system had MAE of 3.98 ± 3.54 days versus 4.17 ± 3.74 days for standard fetal biometry (mean difference −0.21 ± 4.21, 95% CI −0.87, 0.44) which meets the criterion for non-inferiority.

Paired GA estimates for blind sweeps acquired with both a standard ultrasound device and the low-cost device were available for some study participants in the combined test set ($N = 155$ participants). The MAE difference between blind sweeps performed with the low-cost and standard devices was 0.45 days (95% CI, 0.0, 0.9). The mixed effects regression showed that use of the low-cost device was associated with increased error magnitude ($P = 0.001$), although the estimated effect was only 0.67 days.

Fetal malpresentation estimation using blind sweeps acquired with the low-cost ultrasound device were compared against the

**Table 2 Fetal malpresentation estimation.**

| Subset | Number of participants | Number of malpresentations | AUC-ROC (95% CI) | Sensitivity (95% CI) | Specificity (95% CI) |
|---|---|---|---|---|---|
| All | 613 | 65 | 0.977 (0.949, 1.0) | 0.938 (0.848, 0.983) | 0.973 (0.955, 0.985) |
| Low-cost device only | 213 | 29 | 0.970 (0.944, 0.997) | 0.931 (0.772, 0.992) | 0.940 (0.896, 0.970) |
| Standard device only | 598 | 65 | 0.980 (0.953, 1.000) | 0.954 (0.871, 0.990) | 0.977 (0.961, 0.988) |
| Novice only | 189 | 21 | 0.992 (0.983, 1.000) | 1.000 (0.839, 1.000) | 0.952 (0.908, 0.979) |
| Sonographer only | 424 | 43 | 0.972 (0.933, 989) | 0.907 (0.779, 0.974) | 0.987 (0.970, 0.996) |

The fetal malpresentation model was assessed by comparing predictions to the determination of a sonographer. In each subset of the data, we selected only the latest eligible visit from each patient. For sensitivity and specificity computations, model predictions were binarized according to a predefined threshold. Confidence intervals on the area under the receiver operating characteristic (AUC-ROC) were computed using the DeLong method. Confidence intervals on sensitivity and specificity were computed with the Clopper–Pearson method.

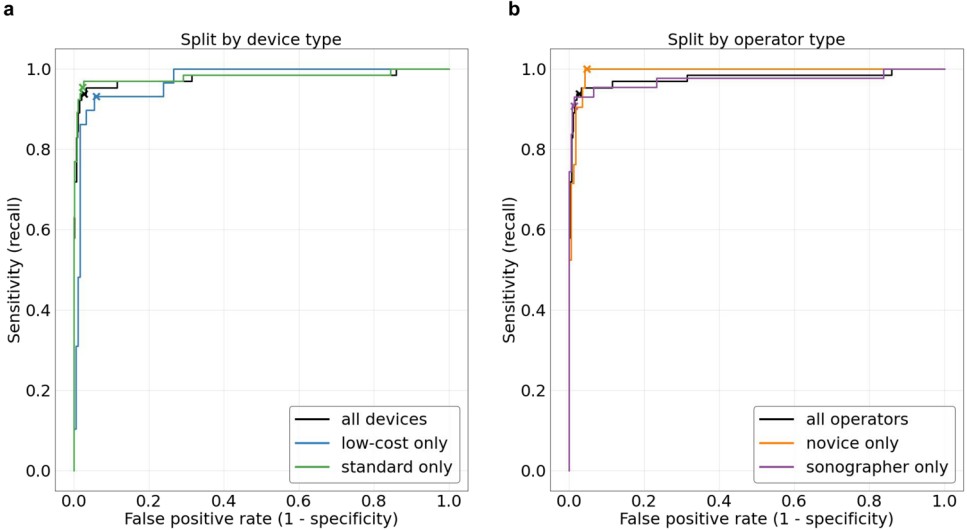

**Fig. 3 Fetal malpresentation estimation.** $n = 623$ study participants. Receiver operating characteristic (ROC) curves for fetal malpresentation estimation. Crosses indicate the predefined operating point selected from the tuning dataset. **a** ROC comparison based on the type of device: low-cost and standard. **b** ROC comparison based on the type of ultrasound operator: novices and sonographers.

sonographer's determination on the combined novice-acquired and sonographer-acquired test sets (213 participants, 29 malpresentations). The blind-sweep AI system had AUC-ROC of 0.97 (95% CI 0.944, 0.997). At the preselected operating point, sensitivity was 0.931 (95% CI 0.772, 0.992) and specificity was 0.94 (95% CI 0.896, 0.970).

**Simplified sweep evaluation**. Protocols consisting of fewer sweeps than the standard 6 sweeps (Fig. 1b) may simplify clinical deployment. We selected M and R sweep types as the best performing set of two sweeps on the tuning set and evaluated this reduced protocol on the test sets.

On test set sweeps performed by sonographers, the reduced protocol of just the M and R sweep types (Fig. 1b) was sufficient for maintaining the non-inferiority of the blind-sweep protocol relative to the standard fetal biometry estimates (MAE difference 95% CI: [−1.5, −0.69] days). The reduced protocol was sufficient for maintaining non-inferiority of blind sweeps relative to standard fetal biometry on test set examinations performed by novices (MAE difference 95% CI: [−1.19, 0.67] days). On average, the reduced protocol can be completed in 20.1 seconds, as extrapolated from videos collected from novices (see

Supplementary Table 2). MAE across subgroups using the reduced protocol are provided in Table 1 (last row).

**Feedback-score evaluation**. Our GA model provided a feedback score to evaluate the suitability of a video sequence for GA estimation. The GA model computed the feedback score for 24-frame video sequences (about one second in length) and therefore provided a semi-continuous feedback signal across the duration of a typical 10-s long blind sweep. The feedback score took the form of an inverse variance estimate and can be used to weight and aggregate GA predictions across blind-sweep video sequences during a study visit. All GA results were computed using this inverse variance weighting aggregation method. More details are provided in "Methods".

As expected, video sequences with high feedback score had low MAE when compared against ground truth GA, and low feedback-score video sequences had high MAE compared against ground truth GA. Figure 2d indicates the calibration of the feedback score on the held-out test datasets. Supplementary Fig. 2c shows example blind-sweep video frames from high and low feedback-score video sequences. The feedback score qualitatively aligns with the degree to which the fetus is visible in the video clip, with the high feedback score left and center-left

**Table 3 Mobile-device model run-time benchmarks.**

| Mobile phone | Processor type | | |
| --- | --- | --- | --- |
| | GPU mean ± standard deviation | CPU w/ XNNPACK library (4 threads) | CPU (4 threads) |
| Pixel 3 | 0.9 ± 0.1 s | 2.1 ± 1.0 s | 13.2 ± 2.9 s |
| Pixel 4 | 0.2 ± 0.1 s | 1.5 ± 0.8 s | 9.8 ± 2.5 s |
| Samsung Galaxy S10 | 0.5 ± 0.1 s | 1.7 ± 1.1 s | 10.3 ± 2.3 s |
| Xiaomi Mi 9 | 1.0 ± 0.2 s | 1.8 ± 1.3 s | 13.7 ± 3.4 s |

Time to model inference results (mean and standard deviation in seconds) measured from the end of a 10-s-long blind-sweep video. Both gestational age and fetal malpresentation models run simultaneously on the same video sequence and image preprocessing operations are included. Near real-time inference is achievable on smartphones with graphics processing units or compute libraries optimized for neural network operations. This enables a simple and fast examination procedure in clinical environments.

examples showing the fetal abdomen and head (respectively). In contrast, the fetus is not visible in the low feedback-score examples (center-right and right).

**Run-time evaluation on mobile phones**. Our blind-sweep AI models were designed to enable near real-time inference on modern mobile phones or standard computers, to enable elimination of waiting time during the clinical examination procedure. We measured both GA and fetal malpresentation model run-time performance using 10-s long blind-sweep videos, which were chosen to match the average length of novice blind sweeps (Supplementary Table 2). Videos were streamed to an Android test application running on Google Pixel 3 and 4, Samsung Galaxy S10, and Xiaomi Mi 9 phones (examples of Android phones that can be purchased refurbished for less than $250 USD). Videos were processed by the GA and fetal malpresentation models simultaneously, with both models executed in the same TensorFlow Lite run-time environment. All necessary image preprocessing operations were also included in the benchmark.

Our results indicated that combined diagnoses for both models are available between 0.2 and 1.0 s on average after the completion of a blind sweep on devices with a graphics processing unit (GPU), and between 1.5 and 2.5 s on average after completion on devices with neural network acceleration libraries for standard CPU processors. See Table 3 for complete benchmark results.

## Discussion

In this study, we demonstrated how our end-to-end blind-sweep mobile-device-optimized AI models can assist novices in LMICs in acquiring blind-sweep ultrasounds to estimate two important obstetric measurements: GA and fetal malpresentation. While there have been multiple GA models proposed in the past, ours is the first to describe an end-to-end system focusing on use in LMIC settings. Three prior studies have described using deep learning on single video frames to either directly estimate GA or estimate head circumference that is then used in fetal biometry formulas[23–25,27]. One prior study describes using the FAMLI dataset to estimate GA through deep learning of blind-sweep ultrasound videos[27]. One prior study describes estimation of fetal malpresentation using AI to first detect fetal anatomy, followed by applying a clinical decision-making matrix[26]. Our models perform as well or better than previously described models. Our GA model estimation was non-inferior to standard fetal biometry estimates and our fetal malpresentation model had high sensitivity and specificity. Both models also had similar performance across sonographer and novice-acquired ultrasounds.

We found that later in pregnancy, there was less deterioration of GA model estimation accuracy compared to the clinical standard fetal biometry. Our models utilize the entire ultrasound video as opposed to only accounting for isolated biometric measurements (e.g., head circumference, femur length..) per the standard fetal biometry. This holistic approach may account for the increased

accuracy later in the pregnancy, when the correlation between GA and physical size of the fetus is less pronounced. This may be especially helpful in providing more accurate estimated GA in LMIC settings where access to ultrasound in early pregnancy is rare[30].

While pairwise comparisons between traditional devices and low-cost devices suggest that traditional devices may perform slightly better, GA model estimation from low-cost devices was non-inferior to standard fetal biometry estimation. This suggests that variation in device performance does not result in clinically significant differences in GA model performance. For our fetal malpresentation model, performance was similar between low-cost and traditional devices.

We focused on improving user experience and simplifying ultrasound acquisition since some of the most vulnerable patients are in geographically remote areas without Internet access. While we initially evaluated on blind sweeps consisting of six sweeps, we found that our GA model performed similarly using only two sweeps. The compatibility of the GA model with this simplified 2-sweep protocol suggests that we may be able to simplify acquisition complexity for novices. Our GA model generates a real-time feedback score that provides information on ultrasound video quality and reliability for use in our AI models. In a clinical setting, this feedback score can potentially notify the ultrasound operator to redo a poorly performed sweep. Both the GA and malpresentation models along with the video quality feedback score have been optimized to run on affordable mobile devices and do not require Internet access.

One limitation of this study is the small sample size, which makes it difficult to evaluate each subgroup individually: novices, sonographers, and ultrasound device type. Our dataset included very few videos for GA less than 12 weeks and greater than 37 weeks so we cannot ensure the AI models generalize for these groups. In addition, we only had a limited number of noncephalic presentations resulting in wide confidence intervals. We plan to validate our findings on a larger cohort to address these limitations. These future studies will also include predictions for other critical maternal fetal diagnostics and pregnancy risk stratification.

While designing the AI models, we addressed obstacles that may be encountered in low-resourced settings where remote care is often delivered through novices with limited training. Overall, tools such as the ones assessed in this study can potentially assist in upleveling the capabilities of both facility and novices in providing more advanced antenatal care. Additionally, the underlying techniques and technology could be applied and studied in other ultrasound-based clinical workflows. The prospective clinical evaluation will be important to evaluate real-world effectiveness, and adaptations may be needed to integrate tools such as this into real-world workflows.

## Data availability

The data used in this study was collected as part of grant OPP1191684 funded by the Bill and Melinda Gates Foundation and is covered by their Global Access program

(https://globalaccess.gatesfoundation.org/). The data will be hosted at synapse.org, and access for research purposes can be requested through the foundation. Source data for Figs. 2 and 3 are available in Supplementary Data 1.

## Code availability

Source code for the artificial intelligence models is provided at https://github.com/Google-Health/google-health/tree/master/fetal_ultrasound_blind_sweeps[31].

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

## Acknowledgements
We would like to thank Yun Liu and Cameron Chen for helpful feedback on the manuscript. This study was partially funded by the Bill and Melinda Gates Foundation (OPP1191684, INV003266). The conclusions and opinions expressed in this article are those of the authors and do not necessarily reflect those of the Bill and Melinda Gates Foundation.

## Author contributions
R.G.G., C.Lee, A.W., and M.S. developed and evaluated the artificial intelligence models. J.S.A.S., B.V., J.T.P., M.P.K., E.M.S., N.S., W.G. III, and B.H.C. developed and executed the FAMLI ultrasound data collection study. J.A.T. and S.M.M. contributed to the interpretation of model evaluation results. R.G.G. and C.C. drafted the manuscript with contributions from C.Lee, A.W., S.S., J.S.A.S., B.H.C., J.A.T., D.T., A.U., K.C., and S.M.M. G.E.D., J.G., and T.Sp. provided technical advice during the development of the artificial intelligence model. C.Lau provided interpretation of ultrasound imagery during model development. T.Sa. and K.L. conducted sonographer and patient experience research during the FAMLI data collection study. T.T. and T.Sa. created the original image elements used in Fig. 1. J.W. and R.P. coordinated collaboration between Google Inc., University of North Carolina, and Bill and Melinda Gates Foundation. S.S., J.S.A.S., D.T., A.U., G.C., L.P., and K.C. established the research goals and direction of the study.

## Competing interests
The authors declare the following competing interests: this study was partially funded by Google Inc. R.G.G., C. Lee, A.W., M.S., J.A.T., S.M.M., C.C., S.S., D.T., A.U., K.C., J.G., G.E.D., T. Sp., T. Sa., K.L., T.T., G.C., L.P., J.W., and R.P. are employees of Google Inc. and own stock as part of the standard employee compensation package. The remaining authors declare no competing interests.
