## [Peer Review File · Communications Medicine]

Reviewers' comments:

Reviewer #1 (Remarks to the Author):

The authors proposed an AI-based system for fetal ultrasound analysis. Specifically, the system can handle automatic GA and fetal malpresentation estimation in videos obtained under the “blind sweep” protocol. Experiments validated on Fetal Age Machine Learning Initiative (FAMLI) and Novice User Study Datasets, the proposed method achieve good GA regression estimation and high AUC-ROC in the binary classification task. Also, the models show robustness across operators (experts and novices) and devices (standard and low-cost). The proposed method seems to have the potential to help the low-to-middle-income countries (LMICs), which lack ultrasound devices for lifesaving treatment. However, I have the following major concerns:

Q1: Title of this paper is not very accurate. It's a very big topic for “AI system for fetal ultrasound”, however, this paper only covers two sub-topics: GA and fetal malpresentation estimation.

Q2: Line 19: INTRODUCTION. The author should summarize the studies about AI-based gestational age (GA) estimation and fetal malpresentation classification, at least including 1-2 sentence descriptions for each work.

Q3: Line 53-58: Blind sweep and standard fetal biometry estimates are two vital concepts in the study. Here, blind sweep procedure has been well introduced. What’s “standard fetal biometry estimates”? The authors should provide a description of it for making the reader easier to follow and understand. Also, is the clinical standard procedure only contain one sweep?

Q4: Line 284-306 : If “M and R were found to be sufficient for maintaining accuracy of gestational age estimation”, why the other four sweeps (L, C1, C2, C3) are required? Also, see Table 1, the MAE performances are 3.8 and 4.0 days, respectively (with very closed variation). Is there a significant difference between performance obtained by using the M&R only and the whole six sweeps? What’s the final setting of AI model (2 or 6)?

Q5: Line 307-318: Caption about “Top left” seems miss.

Q6: Line 325-332: “Blind sweep MAE” with 6 sweeps is the AI based result and “Standard fetal biometry estimates MAE” is based on manual evaluation following clinical standards. Is that right?

a) AI-based method outperforms clinical standard based method, which is good. However, the comparison seems not fair.

b) Similar to Q3, if standard fetal biometry estimates contain only one sweep or one standard plane, it is not fair for comparison. Since blind sweep contain 6 videos, naturally providing more information about GA.

c) Another question is, what are the GA results of blind sweep by clinical-standard-based estimation? Also, what are the GA results of Standard fetal biometry sweep by AI-based estimation?

d) See “Standard fetal biometry estimates MAE \pm sd (days)”, why data collected by novices with standard ultrasound device has better performance than that collected by sonographers (4.8 days vs. 5.2 days)? This seems a little unreasonable.

Q7: Line 343-351: Do different mobile phones have the same GPU type? If not, GPU types including memory, etc., should be provided for better comparison. Also for CPU types.

Q8: Line 452-453: “No intermediate fetal biometric measurements were required during training or generated during inference.” What’s the advantages of this? Fetal biometric measurements can provide additional information for helping the clinical evaluation. Also, during model training, compared to predicting the GA directly, combing the biometric parameters learning through a multi-task manner has the potential to optimize the model training.

Q9: Line 467-469: “For each training set case, fetal malpresentation was specified as one of four possible values by a sonographer (cephalic, breech, transverse, oblique), and dichotomized to “cephalic” vs “non-cephalic”.”

- a) Why the numbers of categories for the position of the fetus are different (four and two) in the description?
- b) If the AI model only requires to focus on binary classification, why “breech, transverse and oblique” annotations are required during labeling?
- c) Besides, if four classification tags have already been labeled by the sonographer, why not consider a four-type classification model?
- d) Further, if learning a four-type classifier is more difficult than a binary model, the author can consider combing the two types of the classification models into one model, which is easy to implement and can help improve the model performance while reducing the learning difficulties.

Q10: Line 483-485: “For the GA model, each blind sweep was divided into multiple video sequences. For the fetal malpresentation model, video sequences corresponded to a single complete blind sweep.” Why the two models have different input sizes (multiple sub-videos and the whole video)? As described, the feature extractor for both MobileNetV2, so the model inputs can theoretically be the same.

Q11: Line 491-492: “The convolutional component uses the MobileNetV2 network as a feature extractor for each video frame.” The authors only perform experiments based on MobileNetV2. However, as a study focusing on mobile device models, the author should consider further experiments about different light-weight architectures, including MobileNetV3 and the Shuffle-Net family, etc. The models can be easily found in Pytorch and Torchvision (also in TensorFlow/Keras).

Q12: Line 537-539: Why the input resolutions of two models are different? The motivation should be well introduced. Also, in line 541-550, motivation of different setting for two models should be provided.

Q13: Line 553-553: “The gestational age regression model uses the gestational age ground truth associated with the case as the training label for all video clips within the case.” An accurate GA value should be calculated and evaluated on the specific plane (i.e., standard plane). However, the authors treat the ground truths of multiple video clips as the same GA. Since not all the planes in the video clips are suitable for GA measurement, this may seriously confuse the model learning.

Q14: Line 562-574: Some unusual parameter settings need to be better explained. For example, “keep probability of 0.863 for the gestational age model”, “the gestational age model began with a learning rate of $4.58e-4$ and ended with $4.58e-7$ after 1 million training steps”, etc.

Q15: Line 588-589: How to ensure all the divided clips have the same length to match the input of LSTM? In other words, the uniform division easily leads to inconsistent sub-video length, how to

handle this?

Q16: Line 602-604: Is fetal presentation model also equip with a LSTM module? If yes, why the whole sweep video can input to the model, without the requirement to strictly match the same length as LSTM like in the GA model? If not, the authors should consider describing the differences between model architectures more clearly.

Q17: Line 618-638: Some vital information is missed on the figures. For example, in part "C", the feedback score of each image should be provided in detail (like putting detailed values on the left-down corner).

Q18: The authors can consider combing the two models into one. This paper focus on mobile model construction, thus, fewer model parameters make deployment easier. The two models have almost the same model architectures (Extended Data Figure 3), so, there is no technical difficulty in merging them together. If works, both training and evaluation will become more simplified, and both GA and classification results can be shown on phone simultaneously, which is faster and memory-saving.

Q19: Typo : The author should double-check the whole text carefully. For example:

Line 97: "an a prior criterion"

Line 112: "bind sweep" -> "blind sweep"

Reviewer #2 (Remarks to the Author):

Part 1 This is a very well written and intriguing manuscript that basically describes the hope for the future. The authors have done an excellent job of both describing the study parameters as well as the results (part 1) While this reviewer does have experience in the techniques described including AI in imaging and imaging education in low resource settings the paper nicely provides evidence that the future is closer than many people think. The data nicely shows the differences between experienced and no experienced, high end machines and more basic machines however to the credit of the study these differences are not extreme.

Specific comments to the authors

- 1.The authors could add a number of other low end innovative scanners that are used by patients that has almost immediate feed back that provides a safety net
- 2.The authors have omitted a number of other references to the 6 step method that has shown value that could be compared to the current study
- 3.Some commercially available low end systems (\$500) that have feedback should also be considered
- 4.What was the reason for less sweeps in the low end group.? It certainly doesn't extend the study time to any significance
- 5.Was there any feed back given by the patient?If so were there specific questions? And if so what was revealed?
- 6.Please define ground truth for the readeers

With regards to reviewing the methods section from line 443 I am confused as to where this fits?Is it really part of the paper? a supplement?

It reads well and is interesting and appears valauble but seems out of place and doesn't flow well

Response to Reviewer #1:

Thank you for your detailed and thoughtful review. Please see our responses to your comments and questions:

The authors proposed an AI-based system for fetal ultrasound analysis. Specifically, the system can handle automatic GA and fetal malpresentation estimation in videos obtained under the “blind sweep” protocol. Experiments validated on Fetal Age Machine Learning Initiative (FAMLI) and Novice User Study Datasets, the proposed method achieve good GA regression estimation and high AUC-ROC in the binary classification task. Also, the models show robustness across operators (experts and novices) and devices (standard and low-cost). The proposed method seems to have the potential to help the low-to-middle-income countries (LMICs), which lack ultrasound devices for lifesaving treatment. However, I have the following major concerns:

- *Q1: Title of this paper is not very accurate. It's a very big topic for “AI system for fetal ultrasound”, however, this paper only covers two sub-topics: GA and fetal malpresentation estimation.*
 - We have revised the title of the manuscript to: “Expanding global access to fetal ultrasound: A mobile-optimized AI system for gestational age and fetal malpresentation assessment”
- *Q2: Line 19: INTRODUCTION. The author should summarize the studies about AI-based gestational age (GA) estimation and fetal malpresentation classification, at least including 1-2 sentence descriptions for each work.*
 - We currently provide a comparison to related work in the Discussion section, first paragraph. To help address the reviewer’s request, we have added (Self et al 2020, ref 25) as an example of determining fetal malpresentation using AI and added a comparison to this work in the Discussion (we weren’t aware of this work at the time the manuscript was first submitted).
- *Q3: Line 53-58: Blind sweep and standard fetal biometry estimates are two vital concepts in the study. Here, blind sweep procedure has been well introduced. What’s “standard fetal biometry estimates”? The authors should provide a description of it for making the reader easier to follow and understand. Also, is the clinical standard procedure only contain one sweep?*
 - The clinical standard doesn’t use sweeps, but instead requires precisely located images of several fetal anatomical structures. We’ve added an explanation (Introduction, third paragraph) of standard fetal biometry estimates and noted the contrast with blind sweeps.
- *Q4: Line 284-306 : If “M and R were found to be sufficient for maintaining accuracy of gestational age estimation”, why the other four sweeps (L, C1, C2, C3) are required? Also, see Table 1, the MAE performances are 3.8 and 4.0 days, respectively (with very closed variation). Is there a significant difference between performance obtained by*

using the M&R only and the whole six sweeps? What's the final setting of AI model (2 or 6)?

- A large number of different types of sweeps were performed during study data collection since it was designed before the AI model results were known. We limited our test set evaluations to 6 and 2 sweeps so that the procedure can be feasibly taught and performed by non-experts in LMIC settings. This is indicated in the "Simplified sweep evaluation" section: either 6 or 2 sweeps are sufficient for maintaining non-inferiority of the sweep procedure relative to the clinical standard, and MAE for both procedures is similar (Table 1). The AI model makes gestational age predictions on fixed length video clips, and these predictions are aggregated together to form the final prediction. The number of sweeps per patient visit changes the number of predictions available for aggregation but the underlying AI model does not change based on the number of sweeps performed. We added clarification to the Mobile Device Inference section, third paragraph.
- Q5: Line 307-318: *Caption about "Top left" seems miss.*
 - We have added the missing caption in Figure 2 of the revised manuscript.
- Q6: Line 325-332: *"Blind sweep MAE" with 6 sweeps is the AI based result and "Standard fetal biometry estimates MAE" is based on manual evaluation following clinical standards. Is that right?*
 - Yes, we added clarification to the caption for Table 1.
- a) *AI-based method outperforms clinical standard based method, which is good. However, the comparison seems not fair.*
 - The primary comparison in the manuscript is (a) AI interpretation of blind sweeps vs (b) the clinical standard of expert sonography, which does not involve any blind sweeps, but rather expert measurement of defined structures.
- b) *Similar to Q3, if standard fetal biometry estimates contain only one sweep or one standard plane, it is not fair for comparison. Since blind sweep contain 6 videos, naturally providing more information about GA.*
 - Fetal biometry does not involve blind sweeps but rather deliberate measurement of expertly-acquired images (HC, AC, FL, etc). It is true that more raw information may be available in the sweeps, but this is a benefit rather than a drawback of using our method. Without AI, there was previously no way to process this information using human interpretation and clinical standard estimates ignore this information.
- c) *Another question is, what are the GA results of blind sweep by clinical-standard-based estimation? Also, what are the GA results of Standard fetal biometry sweep by AI-based estimation?*
 - Clinical standard estimates can not use blind sweeps, they require images carefully selected by an expert sonographer and manual measurement of the size of anatomy in the images. We have clarified this in the Introduction.

- Standard fetal biometry doesn't use sweep videos, but rather single images. An evaluation of AI on these images is an interesting topic for another paper but is out of scope here. This would require a different AI model that operates on single images rather than video.
- *d) See "Standard fetal biometry estimates MAE ± sd (days)", why data collected by novices with standard ultrasound device has better performance than that collected by sonographers (4.8 days vs. 5.2 days)? This seems a little unreasonable.*
 - Standard fetal biometry estimates are always collected on full-size commercial machines by expert sonographers (the procedure cannot be performed by novices) even in the novice dataset. Only the blind sweeps are collected by novices. We have clarified this in the "Fetal Age Machine Learning Initiative (FAMLI) and Novice User Study Datasets" section, first paragraph, and the caption for Table 1. The difference in estimates the reviewer points out is within the standard deviation of random variability and not due to differences in the data collection procedure.
- *Q7: Line 343-351: Do different mobile phones have the same GPU type? If not, GPU types including memory, etc., should be provided for better comparison. Also for CPU types.*
 - We added GPU/CPU/SOC chipset information for the four smartphone devices used for evaluation to the Mobile Device Inference section (last paragraph).
- *Q8: Line 452-453: "No intermediate fetal biometric measurements were required during training or generated during inference." What's the advantages of this? Fetal biometric measurements can provide additional information for helping the clinical evaluation. Also, during model training, compared to predicting the GA directly, combining the biometric parameters learning through a multi-task manner has the potential to optimize the model training.*
 - The purpose of our system is to remove the requirement of an expert sonographer from the equation. An expert sonographer is required to acquire fetal biometric measurements.
 - We mentioned this to emphasize the difference between the way our AI+blind sweep model works (end-to-end from pixels to predictions) versus the clinical standard, which produces intermediate measurements. There may be a potential use case for including fetal biometric measurements during training. However, we experimented with this and in our case it didn't help model performance.
- *Q9: Line 467-469: "For each training set case, fetal malpresentation was specified as one of four possible values by a sonographer (cephalic, breech, transverse, oblique), and dichotomized to "cephalic" vs "non-cephalic"."*
- *a) Why the numbers of categories for the position of the fetus are different (four and two) in the description?*
 - From a clinical perspective, "cephalic" is considered normal while all non-cephalic presentations are abnormal and require further attention. We've added a

clarification in the Algorithm Development section, third paragraph. We envision our AI model serving as a triage tool for flagging abnormal cases for referral.

- *b) If the AI model only requires to focus on binary classification, why “breech, transverse and oblique” annotations are required during labeling?*
 - These labels come from standard sonographic exams during the patient visit, and this is the clinical standard assessment.
- *c) Besides, if four classification tags have already been labeled by the sonographer, why not consider a four-type classification model?*
 - Evaluation metrics for 2-class problems are better defined than multi-class metrics and this 2-class dichotomization is more compatible with the clinical use case for triaging abnormal cases. Multi-class analogues for AUC and sensitivity/specificity are less commonly used and more difficult to interpret.
- *d) Further, if learning a four-type classifier is more difficult than a binary model, the author can consider combining the two types of the classification models into one model, which is easy to implement and can help improve the model performance while reducing the learning difficulties.*
 - While this may be a useful suggestion for future work, it’s not clear that including the additional label information will improve performance of the model when evaluated on the cephalic versus non-cephalic binary task, which we feel is the fundamental problem from a clinical perspective.
- *Q10: Line 483-485: “For the GA model, each blind sweep was divided into multiple video sequences. For the fetal malpresentation model, video sequences corresponded to a single complete blind sweep.” Why the two models have different input sizes (multiple sub-videos and the whole video)? As described, the feature extractor for both MobileNetV2, so the model inputs can theoretically be the same.*
 - The two problems are very different. Gestational age can reasonably be estimated from short video clips (even a single image) and these estimates can be aggregated together to improve performance. Fetal malpresentation can not be determined by looking at a short sequence or individual images. Instead, judging the orientation of the fetus requires viewing the full video to assess spatial relationships between anatomical regions. We have added clarification in the Data Preprocessing section, fourth paragraph.
- *Q11: Line 491-492: “The convolutional component uses the MobileNetV2 network as a feature extractor for each video frame.” The authors only perform experiments based on MobileNetV2. However, as a study focusing on mobile device models, the author should consider further experiments about different light-weight architectures, including MobileNetV3 and the Shuffle-Net family, etc. The models can be easily found in Pytorch and Torchvision (also in TensorFlow/Keras).*
 - We performed experiments with other feature extractors including MobileNetV3 and Resnet during the course of model development and evaluated using our tuning sets. We found MobileNetV2 to offer the best tradeoff between accuracy

and run time speed. We added a clarifying remark in the Model Architecture section, first paragraph.

- *Q12: Line 537-539: Why the input resolutions of two models are different? The motivation should be well introduced. Also, in line 541-550, motivation of different setting for two models should be provided.*
 - This is a technical trade off required for training models on TPU accelerator hardware with finite memory resources. The fetal malpresentation model benefited from longer sequence lengths (see response above about spatial context) but we needed to reduce resolution to accommodate this during training. We added clarification in the Data Preprocessing section, second paragraph.
- *Q13: Line 553-553: “The gestational age regression model uses the gestational age ground truth associated with the case as the training label for all video clips within the case.” An accurate GA value should be calculated and evaluated on the specific plane (i.e., standard plane). However, the authors treat the ground truths of multiple video clips as the same GA. Since not all the planes in the video clips are suitable for GA measurement, this may seriously confuse the model learning.*
 - There’s only one ground truth gestational age for the fetus at the time of visit; there’s no notion of multiple GA values associated with different imaging planes. The model produces an inverse variance confidence score for each clip that is used to downweight the impact of video clips that don’t contain suitable information for GA estimation.
- *Q14: Line 562-574: Some unusual parameter settings need to be better explained. For example, “keep probability of 0.863 for the gestational age model”, “the gestational age model began with a learning rate of 4.58e-4 and ended with 4.58e-7 after 1 million training steps”, etc.*
 - We performed hyperparameter searches based on evaluating the model on the tuning set to determine these parameters. The exact values for the learning rate schedule are determined by hyperparameter search using the Vizier optimization system. We selected the dropout rate via hyperparameter search on the tuning set, also using Vizier. We have added a reference for the Vizier system (ref 36) and a clarifying remark in the Training section, second paragraph.
- *Q15: Line 588-589: How to ensure all the divided clips have the same length to match the input of LSTM? In other words, the uniform division easily leads to inconsistent sub-video length, how to handle this?*
 - The last clip from a video sequence is zero-padded to the required fixed length during both training and testing. Clarification was added to the Data Preprocessing section, third paragraph.
- *Q16: Line 602-604: Is fetal presentation model also equip with a LSTM module? If yes, why the whole sweep video can input to the model, without the requirement to strictly*

match the same length as LSTM like in the GA model? If not, the authors should consider describing the differences between model architectures more clearly.

- Yes, these are fixed length sequences as well. We use more temporal downsampling (skipping between frames) for the fetal presentation model to handle the entire video in one sequence. Since blind sweeps are about 10 seconds long, this allowed handling the full video in a 100 frame long fixed sequence. See remark above on spatial context for the fetal malpresentation model.
- *Q17: Line 618-638: Some vital information is missed on the figures. For example, in part “C”, the feedback score of each image should be provided in detail (like putting detailed values on the left-down corner).*
 - We added clarification indicating these images are selected randomly from the top and bottom feedback score percentiles in the caption for Extended Data Figure 2.
- *Q18: The authors can consider combining the two models into one. This paper focus on mobile model construction, thus, fewer model parameters make deployment easier. The two models have almost the same model architectures (Extended Data Figure 3), so, there is no technical difficulty in merging them together. If works, both training and evaluation will become more simplified, and both GA and classification results can be shown on phone simultaneously, which is faster and memory-saving.*
 - This could be considered for future work, but combining the two models is not simple given the different video pre-processing and sequence lengths used for fetal presentation and GA models. See explanation above on differences between the two estimation problems. We demonstrated the feasibility of executing both models simultaneously on the same device without sacrificing real time performance.
- *Q19: Typo : The author should double-check the whole text carefully. For example: Line 97: “an a prior criterion”.*
 - This reads “an a priori criterion for non-inferiority” and is not a typo. “A priori” indicates the criterion for deciding non-inferiority was determined before analysis was performed.
- *Line 112: “bind sweep” -> “blind sweep”*
 - Corrected in the revised manuscript (Mobile-device-optimized AI gestational age and fetal malpresentation estimation section, third paragraph).

Response to Reviewer #2:

Thank you for your encouraging comments and thoughtful review. Please see our responses below:

Part 1 This is a very well written and intriguing manuscript that basically describes the hope for the future. The authors have done an excellent job of both describing the study parameters as well as the results (part 1) While this reviewer does have experience in the techniques described including AI in imaging and imaging education in low resource settings the paper nicely provides evidence that the future is closer than many people think. The data nicely shows the differences between experienced and no experienced, high end machines and more basic machines however to the credit of the study these differences are not extreme.

Specific comments to the authors

- *1. The authors could add a number of other low end innovative scanners that are used by patients that has almost immediate feed back that provides a safety net*
 - We agree such technology would be helpful if available - though we are not aware of any that have been approved by regulatory authorities to provide clinically relevant user feedback directly to patients. We would be happy to include these in the discussion if the reviewer could provide a few pointers.
- *2. The authors have omitted a number of other references to the 6 step method that has shown value that could be compared to the current study*
 - References 20-26 all relate to using sweep acquisition protocols and/or AI for fetal ultrasound applications. We provide a comparison to these references in the first paragraph of the Discussion section. As suggested by Reviewer 1, we added (Self et al 2020, ref 25) as an example of determining fetal malpresentation using AI and added a comparison to this work in the Discussion (we weren't aware of this work at the time the manuscript was first submitted). We would be happy to cite any additional relevant references.
- *3. Some commercially available low end systems (\$500) that have feedback should also be considered*
 - We are not aware of these systems, could you provide a reference? We used the ButterflyIQ handheld system in our study.
- *4. What was the reason for less sweeps in the low end group.? It certainly doesn't extend the study time to any significance*
 - We conjecture that a two-sweep protocol could reduce the amount of time and complexity required to teach novices to use the system. Our goal in this work was to demonstrate that the system can work with fewer sweeps if necessary. We leave this for future work to assess these tradeoffs more fully in real world settings.
- *5. Was there any feed back given by the patient? If so were there specific questions? And if so what was revealed?*
 - Some patients commented that they enjoyed the additional opportunity to see their baby which was afforded by this study. However, we did not receive systematic feedback that informed the findings discussed in this manuscript.
- *6. Please define ground truth for the readers*
 - Ground truth determination for gestational age and fetal presentation are discussed in the Algorithm Development section (second and third paragraphs)

and a visual description is provided in Figure 1. We added clarification earlier in the section titled Mobile-device-optimized AI gestational age and fetal malpresentation estimation, first paragraph.

- *With regards to reviewing the methods section from line 443 I am confused as to where this fits? Is it really part of the paper? a supplement? It reads well and is interesting and appears valuable but seems out of place and doesn't flow well*
 - This is intended as an Online Methods or Supplement depending on formatting preferences of Communications Medicine. It's intended to provide enough technical detail for AI researchers to reproduce our work, and therefore has a different focus and style than the main article text.

REVIEWERS' COMMENTS:

Reviewer #1 (Remarks to the Author):

Authors have addressed all my comments.

Reviewer #2 (Remarks to the Author):

The authors have done an excellent job at responding to the suggestions and even more impressive in responding to the other reviewer's detailed comments

I would point out that the method used to scan is adapted from

abuhamad A, Zhao Y, Abuhamad S, Sinkovskaya E, Rao R, Kanaan C, Platt L. Standardized six-step approach to the performance of the focused basic obstetric ultrasound examination. *Am J Perinat* 2016; 33 (1): 90-8 5 Aug3 [Epub ahead of print] [PMID: 26238329]

it should be cited

Response to Reviewer 1:

Authors have addressed all my comments.

Thanks again for your thoughtful and thorough review.

Response to Reviewer 2:

*The authors have done an excellent job at responding to the suggestions and even more impressive in responding to the other reviewer's detailed comments
I would point out that the method used to scan is adapted from*

abuhamad A, Zhao Y, Abuhamad S, Sinkovskaya E, Rao R, Kanaan C, Platt L. Standardized six-step approach to the performance of the focused basic obstetric ultrasound examination. Am J Perinat 2016; 33 (1): 90-8 5 Aug3 [Epub ahead of print] [PMID: 26238329]

it should be cited

Thank you for your review, and for providing this helpful reference. We have added it to the final version of our manuscript.